# Bioinformatics Identification of Regulatory Genes and Mechanism Related to Hypoxia-Induced PD-L1 Inhibitor Resistance in Hepatocellular Carcinoma

**DOI:** 10.3390/ijms24108720

**Published:** 2023-05-13

**Authors:** Mohan Huang, Sijun Yang, William Chi Shing Tai, Lingfeng Zhang, Yinuo Zhou, William Chi Shing Cho, Lawrence Wing Chi Chan, Sze Chuen Cesar Wong

**Affiliations:** 1Department of Health Technology and Informatics, The Hong Kong Polytechnic University, Hong Kong SAR, China; 2Department of endocrinology, The First Affiliated Hospital of Wenzhou Medical University, Wenzhou 325000, China; 3Department of Applied Biology and Chemical Technology, The Hong Kong Polytechnic University, Hong Kong SAR, China; 4Department of Clinical Oncology, Queen Elizabeth Hospital, Hong Kong SAR, China

**Keywords:** hepatocellular carcinoma, hypoxia, PD-L1 inhibitor, drug resistance, bioinformatics analysis, molecular target, immune escape, combined treatment

## Abstract

The combination of a PD-L1 inhibitor and an anti-angiogenic agent has become the new reference standard in the first-line treatment of non-excisable hepatocellular carcinoma (HCC) due to the survival advantage, but its objective response rate remains low at 36%. Evidence shows that PD-L1 inhibitor resistance is attributed to hypoxic tumor microenvironment. In this study, we performed bioinformatics analysis to identify genes and the underlying mechanisms that improve the efficacy of PD-L1 inhibition. Two public datasets of gene expression profiles, (1) HCC tumor versus adjacent normal tissue (*N* = 214) and (2) normoxia versus anoxia of HepG2 cells (*N* = 6), were collected from Gene Expression Omnibus (GEO) database. We identified HCC-signature and hypoxia-related genes, using differential expression analysis, and their 52 overlapping genes. Of these 52 genes, 14 PD-L1 regulator genes were further identified through the multiple regression analysis of TCGA-LIHC dataset (*N* = 371), and 10 hub genes were indicated in the protein–protein interaction (PPI) network. It was found that *POLE2*, *GABARAPL1*, *PIK3R1*, *NDC80*, and *TPX2* play critical roles in the response and overall survival in cancer patients under PD-L1 inhibitor treatment. Our study provides new insights and potential biomarkers to enhance the immunotherapeutic role of PD-L1 inhibitors in HCC, which can help in exploring new therapeutic strategies.

## 1. Introduction

Hepatocellular carcinoma (HCC) is one of the most common malignancies with the fourth highest cancer mortality rate in the world, seriously damaging human life and health. Chronic hepatitis B and C viruses, chronic alcohol consumption, and metabolic syndrome are all major clinical risk factors for HCC. Current clinical treatment options for liver cancer are classified into surgical therapies, including liver transplantation, cryoablation, resection, and non-surgical therapies, including chemotherapy, targeted therapy, and immunotherapy [1]. However, eligible treatment approaches become very few for patients in advanced HCC where surgical therapy is not appropriate due to large tumor size, location, number of lesions, and comorbidities [2]. Patients with advanced HCC treated with immune-checkpoint inhibitors (ICIs) reached the objective response rate (ORR) of 36% only with drug combination and even lower than 20% with a single drug [3]. Therefore, novel approaches to clarify the underlying mechanism and enhance the response to immunotherapy are important to improve patient survivability and quality of life.

Hypoxia is a common feature of solid tumors, which is closely associated with poor prognosis. Recent experimental analyses have shown that HCC under a hypoxic environment has significant changes in proliferation, apoptosis, migration, invasion, and epithelial-mesenchymal transition [4]. Therefore, it is important to investigate the molecular mechanisms associated with hypoxia in the HCC microenvironment, and the hypoxia-induced factor (HIF) is the main tumor-adapted transcription factor, consisting of HIF-1α, 2α, and 3α [5]. In the hypoxic microenvironment of solid tumors, high expression of HIF-1α is associated with poor prognosis in various cancers, including HCC [6]. It was shown that in the hypoxic environment, HIF-1α is involved in the hypoxic response and activates hundreds of genes associated with the tumor vasculature and tumor cell adaptation to the hypoxic environment. To activate the HIF-downstream pathways that regulate energy metabolism in tumor cells and the expression of immune checkpoint proteins, HIF can bind to the hypoxia response element (HRE) in the promoter region of genes downstream of HIF [7].

PD-L1 is an important immune checkpoint molecule that primarily regulates cellular apoptosis, and therefore, PD-L1 has an essential impact on tumor growth. An increasing number of studies have found that organs exposed to hypoxic conditions and experimental models of hypoxia showed elevated PD-L1 expression at the affected region [8]. In the hypoxic tumor microenvironment, HIF-1α can upregulate PD-L1 expression. Such PD-L1 expression enhancement can be suppressed by the knockdown of HIF-1α [9]. Thus, PD-L1 may be one of the critical mediators expressed by hypoxic tumor cells. PD-L1 inhibitor combination therapy is currently the first-line treatment option for HCC, but no more than 35% of patients manifested a clinical response [10]. In addition, drug resistance acquired due to PD-L1-mediated immune escape after several years of treatment remains a severe problem for patients with cancer recurrence and metastasis. It has been shown that targeting HIF-1α can eliminate PD-L1-mediated immune escape in the tumor microenvironment and increase immune tolerance in normal tissues [11]. Therefore, we hypothesize that the HIF-1α-stimulated increase in PD-L1 expression is a key factor in drug resistance in hypoxic tumors. 

This study aimed to gain new insights into the mechanisms regulating PD-L1 expression and the PD-L1 immune checkpoint inhibitor resistance in solid HCC tumors in a hypoxic microenvironment. Based on the genomic mechanisms, the potential theranostic molecular biomarkers could be identified so that new therapeutic strategies can be recognized to overcome hypoxia-induced PD-L1 inhibitor resistance. 

## 2. Results

### 2.1. Identification of HSGs and HRGs in HCC

By setting the cutoff values of q and fold change (FC) at 1.139 and 1.128, HSGs constitute 800 genes (400 upregulated and 400 downregulated) that indicate differential expression in GSE14520 between tumor and adjacent non-tumor (Figure 1a). The corresponding heatmap is shown in Appendix A. By setting the cutoff values of q and FC at 1.277 and 1.370, HRGs constitute 800 genes (400 upregulated and 400 downregulated) that indicate differential expression between hypoxic and normoxic environments in GSE41666 (Figure 1b). The corresponding heatmap is shown in Appendix A. The intersection of HCC-signature genes (HSGs) and hypoxia-related genes (HRGs) of the two datasets gave 52 overlapping genes, so-called HCC-Hypoxia Overlaps (HHOs), (Fisher-exact test *p* < 1.047 × 10^−11^), 37 of which were upregulated and 15 of which were downregulated in the hypoxic group compared with the normoxic group in GSE41666 (Figure 1c,d).

### 2.2. Gene Set Enrichment Analyses of HSGs and HRGs

Enrichment analysis of HSGs and HRGs on 202 databases (as of 29 November, 2022) was performed using the gseapy package in Python. Significant enrichment results were found in 190 and 161 databases, respectively. Here, we particularly highlight three groups of databases that gave significant enrichment results closely related to hypoxia, HCC, and PD-L1. 

In the Gene Ontology (GO) databases, HSGs upregulated genes were significantly enriched in “mitosis”, “nucleus”, and “organelles” gene sets, HRGs upregulated genes were mainly enriched in “mitosis”, “spindle”, and “nuclear chromosome” gene sets, HSGs downregulated genes were mainly enriched in “monooxygenase activity“, and HRGs downregulated genes were mainly enriched in “cellular response to decreased oxygen levels ” gene set (Figure 2a,b). The bar chart of expression profile of enriched genes is shown in Appendix A. In the Kyoto Encyclopedia of Genes and Genomes (KEGG) Human databases, upregulated HSGs were enriched in “RNA transport” and “cellular senescence signaling”, “drug metabolism”, “chemical carcinogenesis”, and “apoptosis signaling” pathways (Figure 2d). In the “RNAseq Automatic GEO Signatures Human” database, we found that downregulated HSGs and HRGs together were significantly enriched in the “Rb-immunity downregulating Pd-L1” gene set where *TPX2*, *KIF20A*, *CENPA*, *DLGAP5*, and *LMNB1* were found in the genes in common, whereas upregulated HRGs were significantly enriched in the “Rb-immunity downregulating Pd-L1” and “tissue-resident pancreas Pd-1/Pd-L1” gene sets, respectively (Figure 2c).

### 2.3. Evaluation of the Effect of HHOs on PD-L1 Expression in the TCGA-LIHC Dataset

Multiple regression analysis was used to evaluate the effect of HHOs on PD-L1 expression. Ultimately, 14 genes were identified as relevant risk factors affecting PD-L1 expression and were subsequently used to construct a drug-resistance gene regulator model. The model is represented by a linear combination of regression coefficients multiplied by the relative expression levels of PD-L1 regulator genes, indicating the relative effect of each gene on the drug resistance based on multiple regression analysis.
$$\begin{array}{ll} PD-L1= & 0.076+0.240\times FOS+0.261\times FAM13A+0.443\times DLGAP5\\ & -0.264\times ALDH5A1+0.223\times GABARAPL1\\ & -0.123\times CABYR+0.145\times PIK3R1-0.150\times HGFAC\\ & +0.317\times LMNB1-0.418\times KIF20A-0.434\times TPX2\\ & +0.410\times NDC80+0.121\times EPHA2-0.096\times NEDD4L \end{array}$$

Each of the PD-L1 regulator genes included in the model is associated with a significant effect on drug resistance (*p* < 0.05). It is worth noting that the PD-L1 regulator genes with the top five coefficient magnitudes are *DLGAP5*, *NDC80*, *LMNB1*, *KIF20A*, and *TPX2*. 

### 2.4. Construction of Protein–Protein Interaction Network and Identification of Hub Genes

The 52 HHOs were entered into the STRING database platform for protein–protein interaction (PPI) network analysis, and the results were imported into Cytoscape software to construct the PPI network. Among 52 HHOs, 26 genes had connections with confidence score >0.7 in the PPI network, and thus, the rest were removed from the network representation (Figure 3a). In addition, the degree of node connections was calculated using cytoHubba plug-in, and 10 genes with degree >33 were identified as hub genes, namely, *CCNB1*, *BUB1B*, *KIF4A*, *KIF20A*, *KIF11*, *NDC80*, *TPX2*, *CENPA*, *POLE2*, *DLGAP5* (Figure 3b). 

### 2.5. Survival Analysis and PD-L1 Inhibitor Response Prediction

We used Kaplan Meier Plotter software to plot survival curves for genes, which are the union of PPI hub genes and PD-L1 regulator genes. The results of the survival analysis showed that the expression levels of 15 genes significantly correlated with the poor prognosis of the patients (Appendix A). Particularly, significantly better survival rates after PD-L1 treatment were found in patients with a higher expression level of *NDC80* (HR = 0.76, *p* = 0.024) and *TPX2* (HR = 0.77, *p* = 0.03) than those with a lower expression level (Figure 4a,b). We validated the expression of PPI hub genes and PD-L1 regulator genes in response to PD-L1 treatment. Three of the best performing genes in the sample were *GABARAPL1* (AUC = 0.56, *p* = 0.016), *PIK3R1* (AUC = 0.549, *p* = 0.04), and *POLE2* (AUC = 0.553, *p* = 0.027) (Figure 4c).

### 2.6. KEGG Pathway Enrichment

The KEGG pathway enrichment of PPI hub genes and PD-L1 regulator genes were analyzed using the ShinyGo platform. We found 19 significantly enriched pathways (false discovery rates (FDRs) < 0.032), and most of them are associated with immune cells, inflammatory factors, and apoptosis (Figure 5). It is important to note that two genes, *FOS* (AP-1) and *PIK3R1*, are found in “PD-L1 expression and PD-1 checkpoint pathway in cancer”, while the “endocrine resistance pathway” is activated by hypoxia induction. 

## 3. Discussion

Accumulating evidence suggests that hypoxia has a significant impact on HCC, as hypoxia is a prominent feature of malignancy that not only promotes cancer progression but also poses a challenge to the efficacy of immunotherapy. Therefore, there is an urgent need to find HCC biomarkers associated with hypoxia and immunotherapy and elucidate their linkages. In this study, we identified a total of 52 HHOs, which represent overlapping between HSGs and HRGs derived from the GEO datasets. In gene set enrichment analysis (GSEA), we found that some genes in HSGs and HRGs were closely related to the pathways of PD-L1 expression. Particularly, *TPX2*, *KIF20A*, *CENPA*, *DLGAP5*, and *LMNB1* of HHOs were significantly enriched by the retinoblastoma (RB) immunity downregulating PD-L1 expression pathway. The study identifying this pathway illustrated the tumor suppressor function of hyperphosphorylated RB protein in repressing NF-κB activity and PD-L1 expression [12]. Alternatively, 14 PD-L1 regulator genes were selected from HHOs using regression analysis of a TCGA dataset. Further, 10 hub genes were derived from the PPI network of HHOs. Among PD-L1 regulator genes and PPI hub genes, 4 overlapping genes, *TPX2*, *KIF20A*, *NDC80*, and *DLGAP5*, were found. In a future study, we plan to perform co-immunoprecipitation of these 4 overlapping genes and PD-L1. Therefore, we analyzed the survival and treatment response of PPI hub genes and PD-L1 regulator genes after PD-L1 inhibitor treatment based on clinical data to further validate our in-silico results. *TPX2*, *NDC80*, *POLE2*, *GABARAPL1*, and *PIK3R1* were significantly associated with treatment outcomes. The above-mentioned findings support the crucial roles of *TPX2* and *NDC80* in regulating PD-L1 expression and thus affect the PD-L1 inhibitor treatment outcome.

The phosphoinositide-3-kinase regulatory subunit 1 (*PIK3R1*) gene product is mainly p85α, a regulatory subunit of PI3K enzymes. It is also responsible for splicing the isoforms p55α and p50α, primarily expressed in skeletal muscle and liver [13]. Increasing research has shown that *PIK3R1* plays a vital role in human cancer development. The p85 regulatory subunit of PI3K regulates phosphatidylinositol 3,4,5-triphosphate (PIP3) expression, AKT activation, PTEN phosphorylation, and related mRNA expression, mainly through PI3K pathway, which is involved in tumor growth, apoptosis, and drug resistance [14]. It was found that *PIK3R1* expression was significantly higher in HCC tissues than in adjacent normal tissues [15]. In cell lines of HCC, the knockdown of *PIK3R1* significantly reduced the expression of p-PI3K, p-AKT, and p-mTOR, which were closely associated with the growth and proliferation of tumor cells. Furthermore, in a hypoxic environment, cellular activation of HIF-1α downregulated ROS/PI3K/AKT to adapt to the hypoxic loop, while MAPK also downregulated ROS/PI3K/AKT by reducing ERK1/2 phosphorylation [16].

Targeting protein for Xenopus kinesin-like protein 2 (TPX2) is a microtubule nucleation factor involved in mitotic spindle formation. TPX2 is overexpressed in a variety of malignant tumor tissues, including HCC, colon cancer, breast cancer, esophageal cancer, and cervical cancer. It was found that TPX2 regulates PI3K/AKT/p53/p21 pathway and promotes tumor metastasis and growth in HCC tissues [17]. Downregulation of *TPX2* significantly reduced the expression levels of Bcl-2, c-Myc, and Cyclin D1, inhibited PI3K/AKT signaling, suppressed cell proliferation, and promoted apoptosis, thus possibly preventing the development and progression of HCC [18].

Nuclear division cycle 80 (NDC80/Hec1) is a kinetochore complex protein associated with mitosis and is involved in microtubule binding and spindle assembly [19]. Notably, mutations in NDC80 have been confirmed in the second most prevalent primary liver cancer (cholangiocarcinoma, CCA) [20]. In HCC tissues, NDC80 mRNA expression was significantly higher than that of adjacent tissues and may have a role in reducing apoptosis and promoting HCC development [21]. Interestingly, a few studies have shown that NDC80 is associated with PI3K/AKT, but the core component of its complex, SPC24, is defined to regulate the PI3K/AKT pathway in breast cancer cells and produce oncogenic effects [22].

DNA polymerase epsilon subunit 2 (POLE2) is a DNA polymerase subunit that is involved in the DNA replication process, has DNA repair effects, and reduces the occurrence of mutated genes. POLE2, which potentially acts as a therapeutic target and prognostic factor, is overexpressed in a variety of cancers. One study found that *POLE2* regulates its downstream oncogene STC1, activates AKT phosphorylation, decreases HIF-1α expression levels, and promotes cancer cell proliferation [23].

Gamma-aminobutyric acid (GABA), a receptor-associated protein-like 1 (GABARAPL1), is an autophagosomal protein that is a key PI3K transcriptional target and plays an important role in protein transport, interactions, cell proliferation, and tumorigenesis. GABARAPL1 expression is inversely correlated with cancer metastasis and its high expression is associated with a good prognosis. It has been shown that strong expression of GABARAPL1 attenuates AKT activation, reduces mTOR activation, and increases cancer cell invasion [24,25,26]. In fact, autophagy is a tumor suppressor mechanism, mainly limiting oncogenic stresses, such as DNA damage or oxidative stress, in the early stages of tumorigenesis, but it can also promote the survival of cancer cells under nutrient starvation or hypoxic conditions in the advanced stages of tumor progression.

In summary, 20 genes in union of PD-L1 regulator genes and PPI hub genes are not only differentially expressed in hypoxic HCC tissues but also potentially regulate cancer cells through the PI3K/AKT signaling pathway, according to the above-mentioned evidence. These genes are essential in regulating PD-L1 in hypoxic HCC tumor tissues. They may be potential therapeutic and prognostic biomarkers to enhance the sensitivity of cancer cells to PD-L1 inhibitors and reverse drug resistance. However, the specific regulatory mechanisms among them have not been clarified. We predicted that the related genes of possible regulatory mechanisms would provide new insights into the drug resistance mechanisms of potential genes functioning in hypoxic HCC tissues. TPX2, a critical target of KRAS, has been reported to be involved in the development of pancreatic ductal adenocarcinoma (PDAC) through the regulation of hypoxia-mediated HIF-1 [27]. KRAS is known to be an oncogenic gene that regulates the PI3K/AKT/mTOR signaling pathway [28]. Notably, hypoxia-mediated HIF-1α increases PD-L1 expression in a variety of solid hypoxic tumors via the PTEN/PI3K/AKT signaling pathway, thereby inducing T-cell unresponsiveness or apoptosis [29], which suggests that the PTEN/PI3K/AKT/HIF axes may be an essential part of the occurrence and development of drug resistance mechanisms in hypoxic HCC. Additionally, we found that the potential genes are also interrelated. However, the mechanism of their interactive regulation and the role of inter-regulation with PD-L1 in hypoxic HCC still needs further experimental exploration and validation.

Currently, researchers believe that immunotherapy resistance is induced in tumor cells due to the lack of antigenic mutations, altered antigen processing mechanisms, major histocompatibility complex (MHC) dysfunction, human leukocyte antigen (HLA) expression deficiency, β2 microglobulin (β2M) mutations leading to HLA loss, constitutive PD-L1 expression, loss of T cell function, and altered signaling pathways (PI3K, MAPK, WNT, IFN), but it remains unclear about the holistic molecular mechanism leading to the PD-L1 overexpression, and thus, drug resistance under hypoxic conditions [30].

In previous studies, it was found that PD-L1 expression in tumor cells could be upregulated by interferons (IFN) or cytokines, such as tumor necrosis factor (TNF) [31]. Furthermore, HIF-1α also upregulates TNF expression and increases the absorption of TNF by innate immune cells [32]. This is consistent with our findings that PD-L1 expression is regulated by the upstream pathway activated by TNF. It has been shown that TNFR2 acts as the predominant TNF receptor on activation of CD8+ effector T cells. TNF directly affects CD8+ effector T cells through TNFR2, leading to activation-induced cell apoptosis [33]. The upregulated PD-L1 expression can further lead to the apoptosis of T cells through the interaction with PD-1.

The signaling mechanism of PD-L1 expression could be initiated from the binding of TNF to TNFR1. TNFR1 mutations result in altered mitochondrial function, enhanced oxidative capacity, and mitochondrial reactive oxygen species (ROS) production [34]. Under a hypoxic environment, ROS can regulate the stability of HIF-1α and induce DNA damage, further accumulating the risk of DNA mutation [35,36]. Research indicates that ROS can mediate hypoxia through activation of PI3K/AKT/HIF-1α pathway [37]. In addition, activator protein 1 (AP-1, *FOS* gene) is activated by ROS oxidative stress and is involved in tumor generation and the regulation of vascular endothelial growth factor (VEGF) involved in tumor vasculogenesis [38].

A study explored the possibility of combining immune checkpoint inhibitor treatment with HIF inhibitor to repress tumor progression, enhance anti-tumor immunity, and reduce drug resistance. MK6482 was the first HIF-2α inhibitor approved by the FDA to treat patients with advanced renal cancer. Several chemotherapeutic agents were also used clinically to target HIF expression, such as rapamycin, but they have poor bioavailability at the tumor site (<15%) and poor solubility in water, thus increasing associated therapeutic toxicity [4,39]. Drugs known to repress HIF-1α expression indirectly, such as mTOR inhibitors, can also be used as adjuvant therapy for cancer because it has been shown in preclinical models of HCC that such treatment can suppress tumor growth. HIF-1α inhibitors can not only downregulate the PD-L1 expression in tumors but also upregulate the PD-L1 expression in normal tissues, increasing the tolerance of normal tissues to immunotherapy and reducing adverse events [39]. Therefore, it is essential to explore the mechanisms of HIF-induced PD-L1 resistance in solid hypoxic tumors and develop effective and safe new therapies using a potentially multi-targeted approach. In this comprehensive bioinformatics study, the conceptual signaling mechanism of hypoxia-induced PD-L1 inhibitor resistance has been elucidated (Figure 6). KRAS, NDC80, TPX2, and PIK3R1, which act as molecules intimate to hypoxic stress upstream of PD-L1 in the signaling mechanism, were recognized as the potential targets of agents that could be combined with a PD-L1 inhibitor to overcome drug resistance. Such agents include MD6482, PT2385, ABT-869 (mTOR inhibitor); MD6482 has only been used clinically in patients with advanced renal cancer, while the other drugs are currently only being trialed in preclinical models [35,40]. 

## 4. Materials and Methods

Two datasets, GSE14520 and GSE41666, were downloaded from the GEO database and analyzed to determine hypoxia-induced differentially expressed genes in HCC. Functional enrichment analysis and protein–protein interaction (PPI) network construction screened the differentially expressed genes for hub genes. Multiple regression analysis models were constructed using the expression of common differentially expressed genes in the TCGA database to screen for genes highly associated with PD-L1 expression. Finally, we used the Kaplan Meier plotter to analyze PPI hub genes survival and response rates after treatment with PD-L1 immune checkpoint inhibitors and associated pathway analysis to identify potential pathways regulating PD-L1 expression in the hypoxic HCC tumor microenvironment (Figure 7).

### 4.1. Microarray Data Collection and Processing

Two datasets, GSE14520 and GSE41666, were obtained from the GEO database. In the GSE14520 dataset, a single channel array platform was used to profile the gene expression levels of 214 tumor and 214 paired non-tumor samples of HCC patients. In the GSE41666 dataset, HepG2 HCC cell line samples were exposed to anoxia with 24 h of 0% O_2_ and normoxia with a control of 21% O_2_, respectively. As 3 biological replicates were performed for each condition, the gene expression levels of 6 samples were profiled by the expression beadchip platform.

For each gene interrogated by multiple probes in the microarray chip, the average of expression levels of respective probes was taken to provide an expression matrix of unique gene symbols. The log2-transformation was applied to the expression matrix of the GSE14520 dataset but not to the GSE41666 dataset, which has undergone Variance stabilizing normalization (VSN). Standardization was performed to both matrices to obtain normally distributed expression levels, N (0,1). The flowchart of data processing and analysis is shown in (Figure 8).

### 4.2. Identification of Differentially Expressed Genes

Differential expression analysis was performed based on t-test and fold change (FC). For each gene, the *p*-value generated by t-test indicates the statistical significance of differential expression. To address the issue of multiple hypothesis tests for a huge number of genes, *q*-values, i.e., the estimated false discovery rates (FDRs), were derived from *p*-values based on the Storey-Tibshirani *q*-value procedure [41]. For GSE14520, the related sample *t*-test was used to examine the difference between the tumor and paired non-tumor samples. HCC-signature genes (HSGs) are defined as differentially expressed genes where *q*-value < 0.05, FC > 1.4 (upregulated) and <1/1.34 (downregulated) in HCC tumor compared to paired non-tumor samples. For GSE41666, hypoxia-related genes (HRGs) are defined as the differentially expressed genes with *q*-value < 0.05 and FC > 1.301 (upregulated) and <1/1.199 (downregulated), respectively. The cut-off values of FC were determined based on the quantities of upregulated and downregulated genes to be selected. The Venn diagrams were drawn using the Venny 2.1 platform (https://bioinfogp.cnb.csic.es/tools/venny/, as of 25 November 2022), and the HCC-Hypoxia Overlap (HHO) is defined as the gene set obtained from the intersection of HSGs and HRGs.

### 4.3. GO Function and KEGG Pathway Enrichment Analysis of HSGs and HRGs

In this study, for the HSGs and HRGs, Gene Ontology (GO; http://geneontology.org, as of 29 November 2022) analysis was first performed by Python to obtain the results of HSGs and HRGs enrichment in Biological Process (BP), Cellular Component (CC), and Molecular Function (MF). The results of enrichment in Biological Process (BP), Cellular Component (CC), and Molecular Function (MF) were obtained. Then, the Kyoto Encyclopedia of Genes and Genomes (KEGG; https://www.kegg.jp/, as of 29 November 2022) signaling pathway enrichment analysis was performed, and the pathway enrichment results were obtained. The adjusted *p*-value < 0.05 and FDR adjusted *p*-value < 0.05 were statistically significant and were the thresholds for selecting the major enrichment functions and pathways of the HSGs and HRGs.

### 4.4. Gene Set Enrichment Analysis (GSEA) of HSGs and HRGs

GSEA is a computational method for analyzing and interpreting changes in gene pathway levels and association analysis in transcriptomics experiments, including genome-wide association studies and RNA-seq gene expression experiments. The random permutation procedure (permutation_num = 1000) using the gseapy-v1.0.0 python library was used to obtain the zero distribution. The Enrichr method in the gseapy-v1.0 python library was used to determine the signaling pathways regulated by hypoxia-related features of HCC (adjust *p* value < 0.05). The gseapy-v1.0.0 package currently supports a library of 202 databases. 

### 4.5. PPI Network Construction and Identification of Hub Genes

STRING (Search Tool for the Retrieval of Interacting Genes, http://string-db.org/, as of 12 December 2022) is an online database for searching protein interactions. To further explore the interactions among HHOs, which represent the overlapping genes between HSGs and HRGs, the HHOs were imported into STRING to obtain the PPI network with a confidence score > 0.7 [42]. Among the HHOs, the PPI hub genes were identified with a degree threshold using the degree algorithm of cytoHubba, where the connectivity degree of a gene is defined as its connected neighbors [43]. A threshold was selected to include 10 PPI hub genes with the top degree values for further analysis.

### 4.6. Multiple Regression Analysis of the Effect of HHOs on PD-L1

To examine the effect of HHO on PD-L1 expression, we used multiple regression analysis. RNAseq expressions of TCGA-LIHC were obtained from the UCSC xena website (https://xenabrowser.net/datapages/ as of 6 December 2022), including 371 HCC tissue samples, and RNAseq expressions of HHOs and PD-L1 were extracted and standardized. Compared with machine learning models, multiple regression could produce more stabilized and re-producible results without fixing a particular random seed. We adopted stepwise forward algorithm (*p* value < 0.05) to select the genes from HHO that significantly and substantially affect the PD-L1 expression level, denoted by Y in the following formula:Y = b_0_ + b_1_X_1_ + b_2_X_2_ + … + b_n_X_n_,
where X_n_ represents the expression level of the nth selected gene and b_n_ represents the corresponding coefficient quantifying its effect on PD-L1 expression. The selected genes are denoted by PD-L1 regulator genes.

### 4.7. Survival Analysis and PD-L1 Inhibitor Response Prediction

Kaplan Meier plotter (KM plotter; http://kmplot.com/analysis/ as of 21 December 2022) is a survival analysis platform containing clinical data and gene expression data with survival information from GEO, EGA, and TCGA databases. We plotted survival curves and calculated risk ratios for log-rank *p* values and 95% confidence intervals for each of PPI hub genes and PD-L1 regulator genes in cancer patients. The patients were stratified into higher and lower expression groups based on the median expression level of each gene before PD-L1 inhibitor treatment.

The ROC plotter (https://www.rocplot.org/ as of 23 December 2022) is a tool that enables the identification of predictive biomarkers based on gene expression using transcriptomic data from many cancer patients. It was used to evaluate the ability of expression level of each of the PPI hub genes and PD-L1 regulator genes in predicting the response to the PD-L1 inhibitor based on 454 pan-cancer patients in the database.

In addition, we performed KEGG pathway enrichment analysis with the criteria of FDR < 0.05 for the union of PPI hub genes and PD-L1 regulator genes in the ShinyGo 0.76.3 platform (http://bioinformatics.sdstate.edu/go/ as of 1 January 2023).

## 5. Conclusions

Overall, a comprehensive bioinformatics analysis of hypoxia-induced PD-L1 inhibitor resistance in HCC was performed and revealed that genes such as *TPX2*, *NDC80*, *POLE2*, *GABARAPL1*, and *PIK3R1* may be involved in the PI3K-AKT signaling pathway and play an essential role in the pathological and physiological processes of hypoxia-induced PD-L1 inhibitor resistance. The results of this study may provide potential therapeutic targets and deepen the understanding of the underlying mechanisms of hypoxia-induced PD-L1 inhibitor resistance.

## Figures and Tables

**Figure 1 ijms-24-08720-f001:**
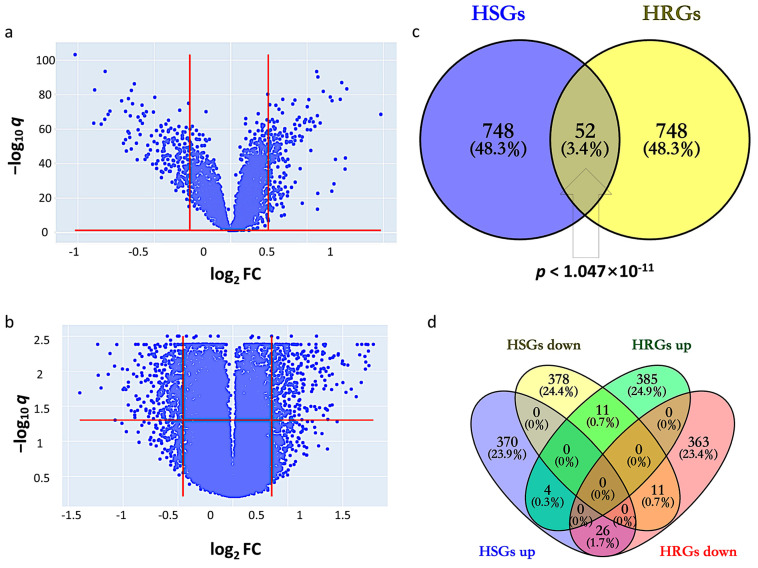
Identification of HCC-signature genes (HSGs) and hypoxia-related genes (HRGs). (**a**) Volcano plot for GSE14520; (**b**) Volcano plot for GSE41666; (**c**) Overlapping genes between HSGs and HRGs; (**d**) Overlapping genes among upregulated and downregulated genes in HSGs and HRGs.

**Figure 2 ijms-24-08720-f002:**
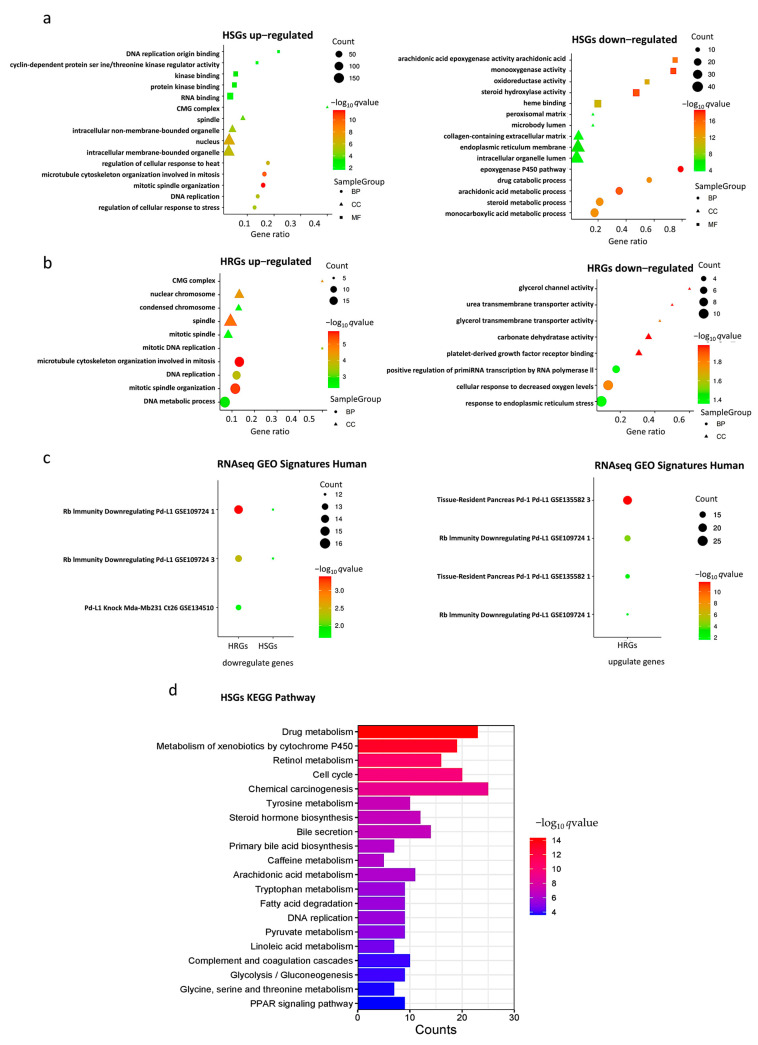
Gene set enrichment analysis. (**a**) Gene Ontology (GO) enrichment analysis of HSGs; (**b**) GO enrichment analysis of HRGs; (**c**) RNAseq GEO Signatures Human enrichment analysis of HSGs and HRGs; (**d**) Kyoto Encyclopedia of Genes and Genomes (KEGG) enrichment analyses of HSGs.

**Figure 3 ijms-24-08720-f003:**
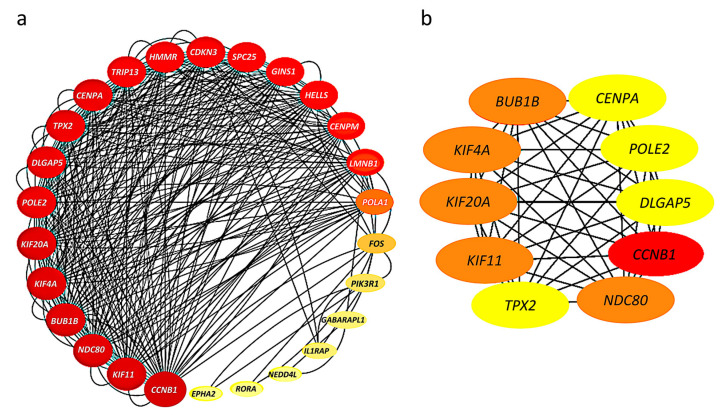
Protein–protein interaction (PPI) networks: (**a**) Connections among 26 genes with confidence score >0.7; The larger the node, the greater the degree of connectivity and the darker the color, the greater the combined score value; (**b**) 10 hub genes with connectivity degree >33; Genes with a confidence score ≤0.7 are not shown here; The darker the color, the stronger the degree of criticality.

**Figure 4 ijms-24-08720-f004:**
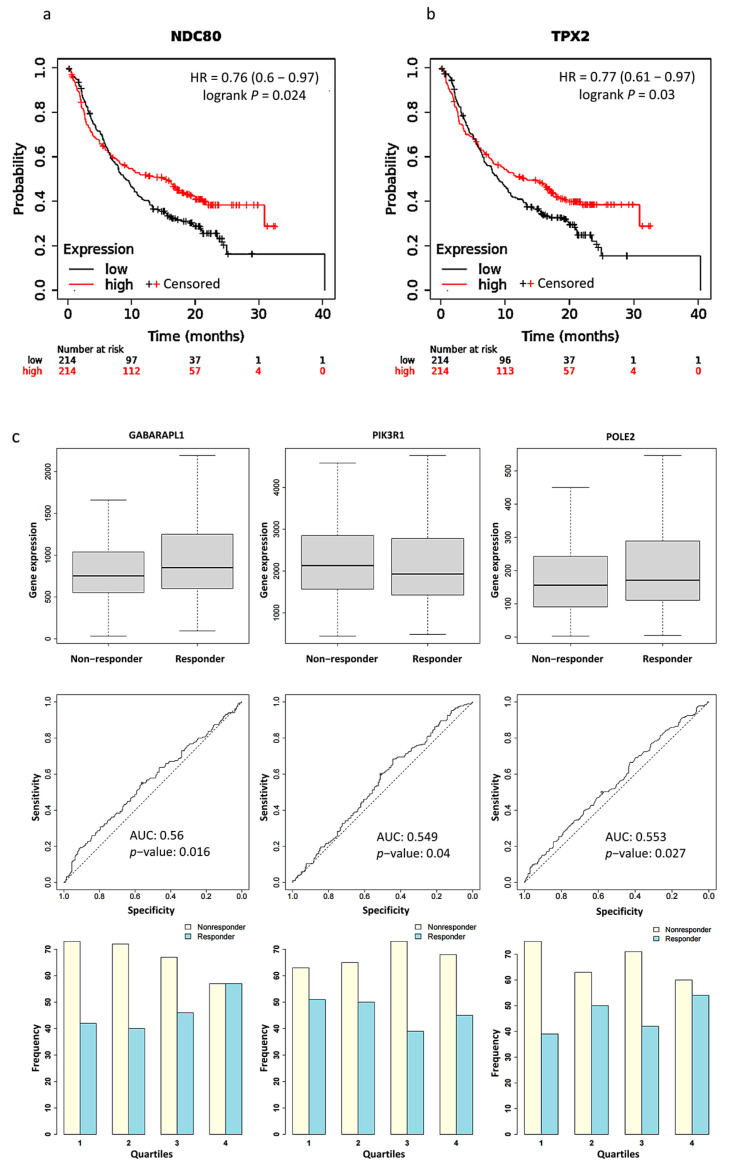
Kaplan–Meier plots for the comparison of survival between high and low expression levels: (**a**) *NDC80* (HR = 0.76, *p* = 0.024); (**b**) *TPX2* (HR = 0.77, *p* = 0.03). (**c**) Boxplots, Receiver operating characteristic (ROC) curves, and responders’ frequency of top three genes in predicting PD-L1 inhibitor response: *GABARAPL1* (AUC = 0.560, *p* = 0.016), *PIK3R1* (AUC = 0.549, *p* = 0.04), and *POLE2* (AUC = 0.553, *p* = 0.027); “^o^” indicates the strongest cutoff, which has the minimal distance from the ideal discriminator.

**Figure 5 ijms-24-08720-f005:**
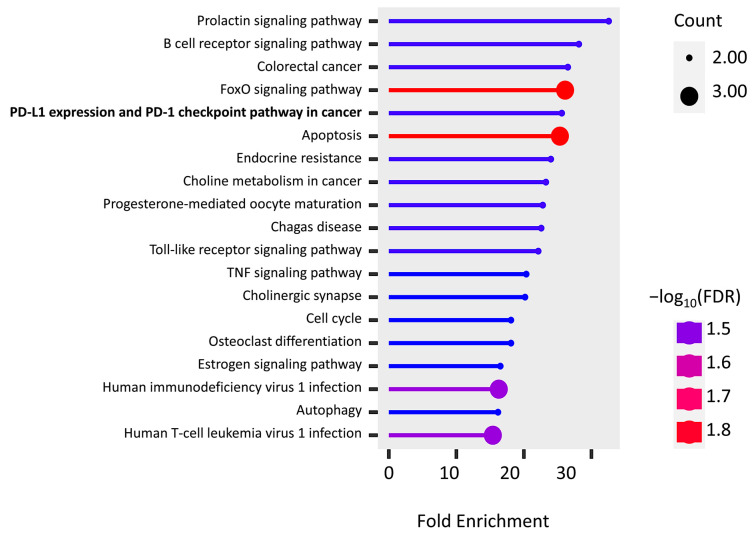
KEGG pathway enrichment of the union of PPI hub genes and PD-L1 regulator genes.

**Figure 6 ijms-24-08720-f006:**
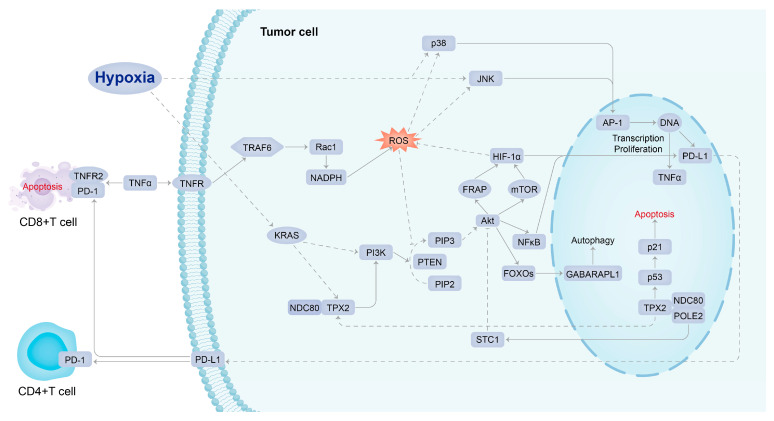
Conceptual signaling mechanism of hypoxia-induced PD-L1 inhibitor resistance.

**Figure 7 ijms-24-08720-f007:**
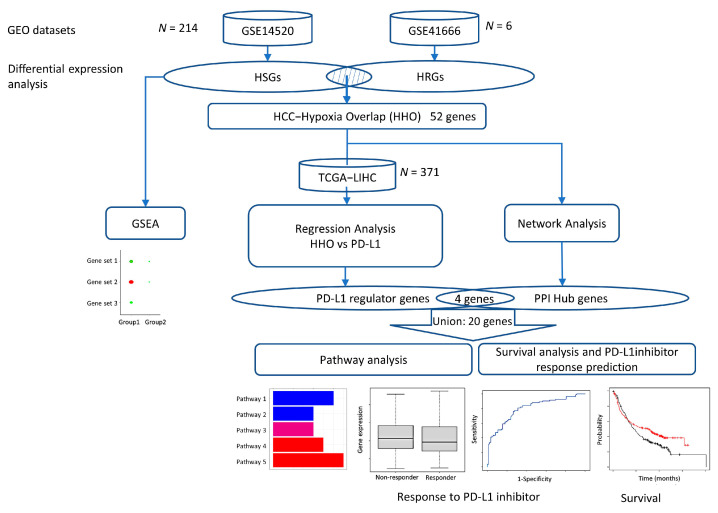
Flowchart of this study.

**Figure 8 ijms-24-08720-f008:**
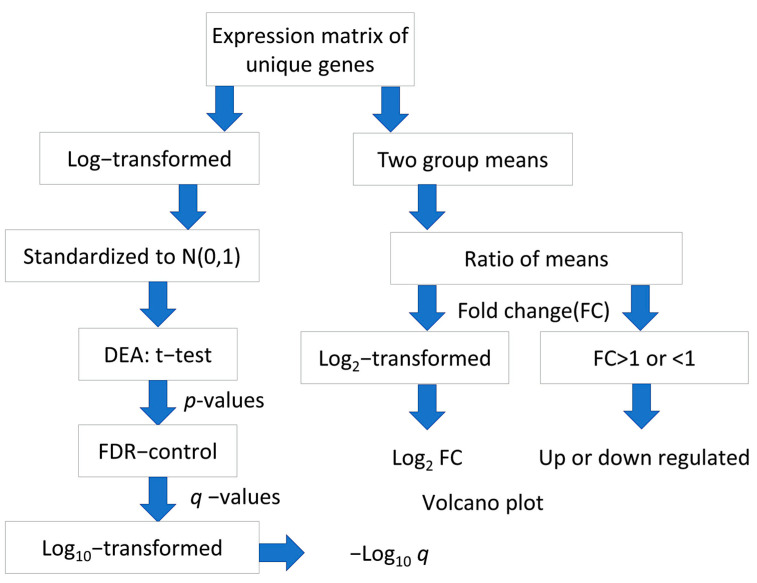
Flowchart of processing and analysis of microarray datasets.

## Data Availability

https://www.ncbi.nlm.nih.gov/geo/query/acc.cgi?acc=gse14520; https://www.ncbi.nlm.nih.gov/geo/query/acc.cgi?acc=GSE41666; https://portal.gdc/cancer.gov/projects/TCGA-LIHC.

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
