# Peer review of "Bioinformatics Identification of Regulatory Genes and Mechanism Related to Hypoxia-Induced PD-L1 Inhibitor Resistance in Hepatocellular Carcinoma"

_ijms, 2023, doi:10.3390/ijms24108720_

Round 1
Reviewer 1 Report
The abstract gives a short but informative overview and serves as a well constructed introduction to the paper.
The paper is generally well written and gives a precise report of the analysis.
Concerning the structure, based on personal preference, I feel it makes more sense to mention materials and methods before results.
The main issues concern the figures and graphics:
1) many (e.g. 1, 2, 4) are to small and hard to read.
2) Figure 6 is hard to follow and seems a little complicated. Is there maybe an animated version of the pathway available?
Author Response
Dear Editor and reviewers,
We would like to express our sincerely thanks to the comments of editor and reviewers and the chance for major revision of our manuscript.
Our responses to the comments are given as follows.
Reviewer 1
1.1 Concerning the structure, based on personal preference, I feel it makes more sense to mention materials and methods before results.
Response: We prefer to follow the standard paper structure of IJMS.
1.2 The main issues concern the figures and graphics:
1) many (e.g. 1, 2, 4) are to small and hard to read.
Response: The texts have been enlarged in figures 1-6 and the text colour has been changed to higher contrast in figure 3 so that they are easy to read.
2) Figure 6 is hard to follow and seems a little complicated. Is there maybe an animated version of the pathway available?
Response: Figure 6 has been changed to an animated version of the pathway so that the illustration of cell membrane, nuclear membrane, CD8+ T cell and CD4+ T cell becomes more vivid.
We found that the labels of normoxia and anoxia were mistakenly swapped. Figure 1b, 1d has been corrected. The other results are not affected after checking.
All the revisions to the manuscript have been marked up using the “Track Changes” function.
We confirm that all references are relevant to the contents of the manuscript.
Thank you for your consideration of our work.
Best Regards,
Dr. Lawrence Wing-Chi CHAN Ph.D.
Associate Professor
Department of Health Technology and Informatics
Hong Kong Polytechnic University
Tel: (852) 3400 8561
Fax: (852) 2362 4365
Reviewer 2 Report
Huang et al. discussed that the combination of PD-L1 inhibitor and the anti-angiogenic agent had become the new reference standard for hepatocellular carcinoma (HCC) due to the survival advantage. Still, its objective response rate remains low. The authors described evidence that PD-L1 inhibitor resistance is attributed to the hypoxic tumor microenvironment. In this study, the authors did bioinformatics analysis to identify genes and the underlying mechanisms that improve the efficacy of PD-L1 inhibition. The study will interest the readers, but I have some concerns.
1. The quality of the images is not good. It is not clear.
2. The authors claimed " HSGs up-regulated genes were significantly 104 enriched in “mitosis”, “nucleus”, and “organelles” gene sets, and HRGs up-regulated 105 genes were mainly enriched in “mitosis”, “spindle”, and “nuclear chromosome” gene sets 106 (Fig. 2a,b)"- Please show the FPKM values and the bar charts for the essential genes, such as the cell cycle markers.
3. Please make a heat map for the HSG and HRG also.
4. The authors found some important PPI in Fig 3. Do the authors have a plan to confirm these using co-immunoprecipitation for future studies?
5. Please improve Figure 5 for the KEGG pathway.
Author Response
Dear Editor and reviewers,
We would like to express our sincerely thanks to the comments of editor and reviewers and the chance for major revision of our manuscript.
Our responses to the comments are given as follows.
Reviewer 2
2.1 The quality of the images is not good. It is not clear.
Response: The sharpness has been raised in figures 1-8 and the text colour has been changed to higher contrast in figure 3 so that they become clearer.
2.2 The authors claimed " HSGs up-regulated genes were significantly 104 enriched in “mitosis”, “nucleus”, and “organelles” gene sets, and HRGs up-regulated 105 genes were mainly enriched in “mitosis”, “spindle”, and “nuclear chromosome” gene sets 106 (Fig. 2a,b)"- Please show the FPKM values and the bar charts for the essential genes, such as the cell cycle markers.
Response: We have made Figure S3, showing a bar chart of expression profiles of HRGs up-regulated in gene set “microtubule cytoskeleton organization involved in mitosis”. Instead of FPKM values, gene expression levels are shown because microarray data is used.
2.3 Please make a heat map for the HSG and HRG also.
Response: We have made Figure S1, showing the heatmap for HSGs, and Figure S2, showing the heatmap for HRGs.
2.4 The authors found some important PPI in Fig 3. Do the authors have a plan to confirm these using co-immunoprecipitation for future studies?
Response: Yes, we have a plan to perform co-immunoprecipitation of these 4 overlapping genes and PD-L1. A sentence has been added at lines 218-220.
2.5 Please improve Figure 5 for the KEGG pathway.
Response: The texts have been enlarged and the font of “PD-L1 expression and PD-1 checkpoint pathway in cancer” has been changed to bold in Figure 5.
We found that the labels of normoxia and anoxia were mistakenly swapped. Figure 1b, 1d has been corrected. The other results are not affected after checking.
All the revisions to the manuscript have been marked up using the “Track Changes” function.
We confirm that all references are relevant to the contents of the manuscript.
Thank you for your consideration of our work.
Best Regards,
Dr. Lawrence Wing-Chi CHAN Ph.D.
Associate Professor
Department of Health Technology and Informatics
Hong Kong Polytechnic University
Tel: (852) 3400 8561
Fax: (852) 2362 4365
Round 2
Reviewer 2 Report
I have no further comments.